# Digital transformation of an academic hospital department: A case study on strategic planning using the balanced scorecard

Thomas Hügle [1]*, Vincent Grek[1,2]

**1** Department of Rheumatology, Departement Appareil Locomoteur (DAL), University Hospital Lausanne (CHUV) and University of Lausanne, Switzerland, **2** Department of Urology, Inselspital and University of Bern, Bern, Switzerland

* Thomas.hugle@chuv.ch

**Data Availability Statement:** All relevant data are included in the manuscript.

**Funding:** The authors received no specific funding for this work.

## Abstract

Digital transformation has a significant impact on efficiency and quality in hospitals. New solutions can support the management of data overload and the shortage of qualified staff. However, the timely and effective integration of these new digital tools in the healthcare setting poses challenges and requires guidance. The balanced scorecard (BSC) is a managerial method used to translate new strategies into action and measure their impact in an institution, going beyond financial values. This framework enables quicker operational adjustments and enhances awareness of real-time performance from multiple perspectives, including customers, internal procedures, and the learning organization. The aim of this study was to adapt the BSC to the evolving digital healthcare environment, encompassing factors like the recent pandemic, new technologies such as artificial intelligence, legislation, and user preferences. A strategic mapping with identification of corresponding key performance indicators was performed. To achieve this, we employed a qualitative research approach involving retreats, interdisciplinary working groups, and semi-structured interviews with different stakeholders (administrative, clinical, computer scientists) in a rheumatology department. These inputs served as the basis for customizing the BSC according to upcoming or already implemented solutions and to define actionable, cross-level performance indicators for all perspectives. Our defined values include quality of care, patient empowerment, employee satisfaction, sustainability and innovation. We also identified substantial changes in our internal processes, with the electronic medical record (EMR) emerging as a central element for vertical and horizontal digitalization. This includes integrating patient-reported outcomes, disease-specific digital biomarker, prediction algorithms to increase the quality of care as well as advanced language models in order save resources. Gaps in communication and collaboration between medical departments have been identified as a main target for new digital solutions, especially in patients with more than one disorder. From a learning institution's perspective, digital literacy among patients and healthcare professionals emerges as a crucial lever for successful implementation of internal processes. In conclusion, the BSC is a helpful tool for guiding digitalization in hospitals as a

**Competing interests:** I have read the journal's policy and the authors of this manuscript have the following competing interests: TH has received speaker fees or research grants from Fresenius Kabi, Abbvie, BMS, Lilly, Janssen and GSK. He is a scientific board member of Atreon and Vtuls and patent holder of a digital biomarker for joint swelling. VG has no competing interest.

horizontally and vertically connected process that affects all stakeholders. Future studies should include empirical analyses and explore correlations between variables and above all input and user experience from patients.

## Author summary

Digital transformation enhances hospital efficiency and quality, yet the integration of these technologies poses challenges that require clear direction. The Balanced Scorecard (BSC) is a tool that helps institutions gauge and action new strategies, not limited to financial metrics. It promotes rapid adjustments and offers clarity on performance on different perspectives. The university hospital sector is suitable for the application of the BSC, as the financial perspective is important, but other perspectives such as patient care and safety or research and innovation are equally important. This study adapted the BSC for current and future digital health solutions such as digital therapeutics or biomarker, AI or automation. Key performance indicators span across education, employee satisfaction and patient empowerment. By collecting insights from diverse stakeholders at a Swiss University Hospital, we developed a custom BSC. This updated BSC accentuates the role of electronic medical records in digitalization and underscores communication challenges between departments. A crucial insight is the importance of digital health literacy for both patients and staff. In essence, the BSC adeptly steers hospitals through digital transitions. Future studies should emphasize real-world testing and patient feedback.

## Introduction

Digital transformation is an ongoing process in hospitals that has enormous potential to improve patient care, optimize costs, and streamline resources [1]. The advancements in computing power, data storage, and interoperability, such as the electronic medical records (EMR) with mobile devices, and the growing availability of artificial intelligence (AI) to harness this data flood, are reshaping healthcare [2]. These changes are occurring against the backdrop of an aging society with more comorbidities, increasing the need for interdisciplinary collaboration. However, healthcare systems are also encountering rising costs due to advanced diagnostic and therapeutic interventions and a critical shortage of healthcare professionals. Therefore, it is essential to reorganize healthcare at the point of care as well as on institutional and systemic levels using new digital tools, including AI and automation [3].

Integrating new digital solutions in hospitals is challenging and requires breaking down data and knowledge silos. On a vertical level, healthcare professionals, administrative staff, and patients use different indicators to measure quality and satisfaction and often do not speak the same language. The same is true horizontally, with a considerable lack of communication and data usage both between medical specialties and patients. This notably applies for overlapping pathologies such as immune-mediated disorders or chronic pain requiring parallel care. Hospitals hardly use digital devices to overcome these silos, apart from using the same EMR. Interdisciplinary case discussions in person, by email, or phone are rare occasions to interact with colleagues from other disciplines to take decisions and learn from each other. However, due to the lack of time and staff, such meta-networks are hardly scalable.

A range of new digital solutions addresses these issues. Many are already certified, such as >500 FDA-cleared AI algorithms, certified mobile health applications, and digital therapeutics such as the German DIGAs (Digitale Gesundheitsanwendung). Other solutions with a

potentially large impact are upcoming, such as large language models (LLM) and transfer learning models that exploit and harmonize unstructured clinical data and biomarkers across disciplines. New applications are also available for the EMR as a central digital element in patient care, expanding and leveraging it horizontally and vertically, such as AI or mobile patient applications.

Responsible persons in hospitals are aware of this development, but the implementation of digital tools in clinical practice is anything but easy. Digital transformation requires a high investment in technology and stakeholder training to create gains in operational efficiency, medical care, and cost-reduction. Different expectations from the economic, clinical, and patient perspective make it difficult to prioritize and approach digitalization implementation.

This study analyzes existing and emerging needs of our hospital department and projects existing and new digital solutions across the value chain. The article presents a systematic methodology that takes into account the mission and vision of the institution, current health-care trends, and shows the impact of digital tools on key processes and their indicators. A central element of this study is the Balanced Scorecard (BSC) as a strategic management and planning system to improve internal operations and project external outcomes [4]. It has been developed to monitor and improve real-time performances, for operational adjustments and for implementation of new strategies [5]. The BSC is adapted to the hospital perspective with a focus on clinical, patient, quality of care, and innovation outcomes and their key performance indicators (KPIs) [6]. The possible impact of existing and upcoming digital solutions on these outcomes is analyzed, and internal processes and knowledge are discussed to implement them in clinical practice.

## Methods

### Data

This monocentric, observational study on strategic planning was conducted in the rheumatology department of the University Hospital in Lausanne (CHUV), Switzerland from 2017–2023. It included restrictions and subsequent new digital developments during the SARS-Cov2 pandemic. Data were used from a general retreat, working and focus groups as well as semi-structured interviews with different stakeholders and a literature and social media review. The methodological process is illustrated in Fig 1.

### Retreat and working groups

An initital retreat with all institutional stakeholders on a cadre level (doctors, nurses, adminsitrative) was performed 2017 to define visions, missions and clinical priorities. A SOAR analysis (Strengths, Opportunities, Aspirations, and Results) has been performed [7]. Clinical needs and potential digital solutions were further elaborated in monthly working groups over six months. Subsequently, existing key processes were analyzed in four working groups (patient care, organisation, research and education).

### Semi-structured interviews, focus groups and conferences on an institutional level

We performed interviews with all stakeholders, including the informatic department, administration, clinicians, nurses and researchers. The status quo of implemented digital solutions and ongoing projects in terms of digital transformation were investigated. Data access and interfaces on an institutional level were discussed. From a clinical side, we organised a yearly inter-disciplinary conference between rheumatology, gastro-enterology and dermatology on

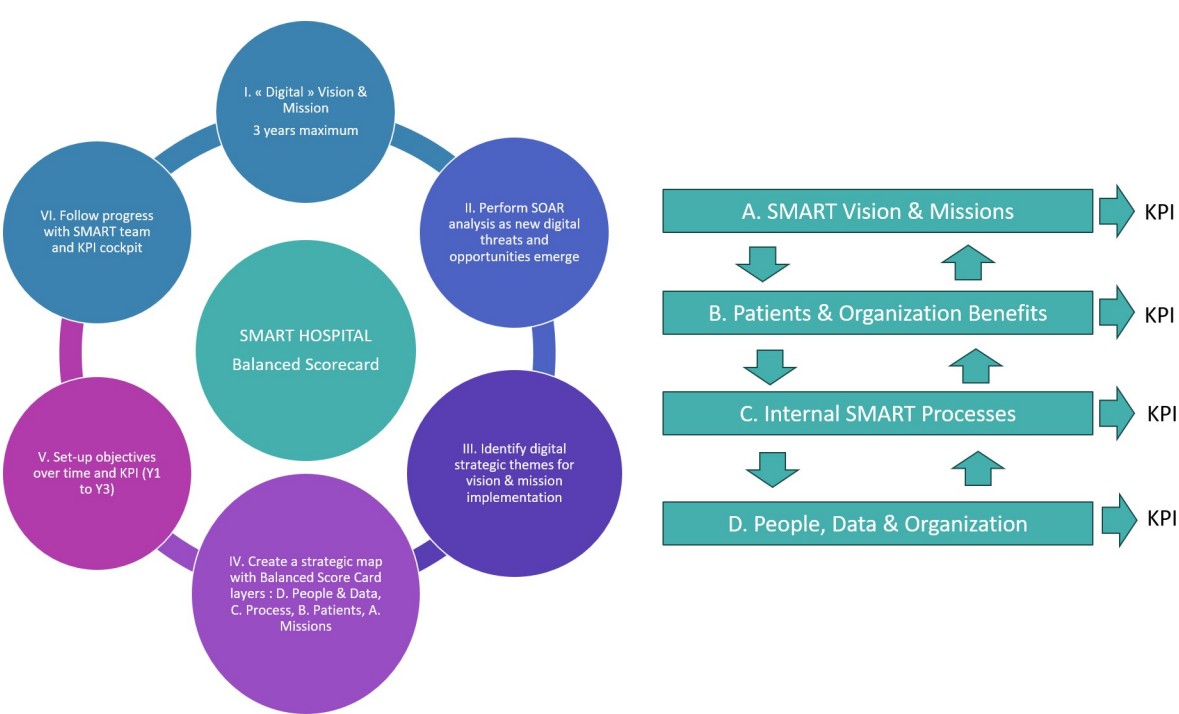

**Fig 1. Methological view on the implementation of the Balanced Scorecard.**

immune-mediated diseases to understand the clinical needs and developments of each speciality (www.common-ground-meeting.org).

## Technological review

Information on existing digital solutions for health care insititutions was obtained by literature review on Pubmed, Google, social media (mainly LinkedIn) and podcasts (e.g. Medical AI Podcast, DTx Podcast, Faces of Digital Health, Deep Minds). We created the « Digital Rheumatology Network » (www.digitalrheumatology.org) as international platform to educate digitalization in rheumatology along with a conference series called Digital Rheumatology Day was started 2019. This platform served inform and connect clinicians and researcher with digital companies and start-ups in the field.

## Adapted Balanced Scorecard and Strategy Mapping

The Balanced Scorecard was adapted to measure or project performances of digital solutions according four perspectives: financial, customer, internal processes and learning and growth. Vision, mission and have been elaborated in the working groups. Internal and external customers have been defined as patients, clinicians, health care professionals and administration. Financial performance has been shifted to « values », which were defined as: patient empowerment, clinical-decision support, time-saving, cost-effectiveness and quality.

## Variables and KPI

KPI have been identified and assessed for measurability and feasibility in the working groups based on a clinical, internal, learning, quality of care or innovation perspective. According the clinical focus of this work KPIs shown in Fig 1 concern efficiency and effectiveness, qualiy of

care, time management, scientific development of health care professionals, patient-centered-ness, technology and information systems and interdisciplinary communication.

# Results

## Strategic mapping

The primary result of this work is a strategic map based on an adapted Balanced Scorecard according key procedures for digitalization (Fig 2). Previously, missions, visions and values facing new medical trends have been elaborated in a retreate and subsequent working groups. Internal processes and knowledge in terms of digitalisation have been worked out in semi-structured interviews, literature and social media search. Each chosen subset of the BSC was aligned with current developments in healthcare and digital technology.

## Mission and vision

We defined the mission of our academic department as cost-effective and state-of-the-art patient care, combined with a high level education and research. The vision was to overcome the challenges that come along with demographic changes of our society and staff shortage by becoming a smart hospital. We envision a countinously fluctuating healthcare system, e.g. due to the occurrence of infection waves, accidents or natural phenomena (heat or cold waves etc.). We aim to foster digital transformation but also to celebrate human interaction and contact with patients that are in control of their decisions and data.

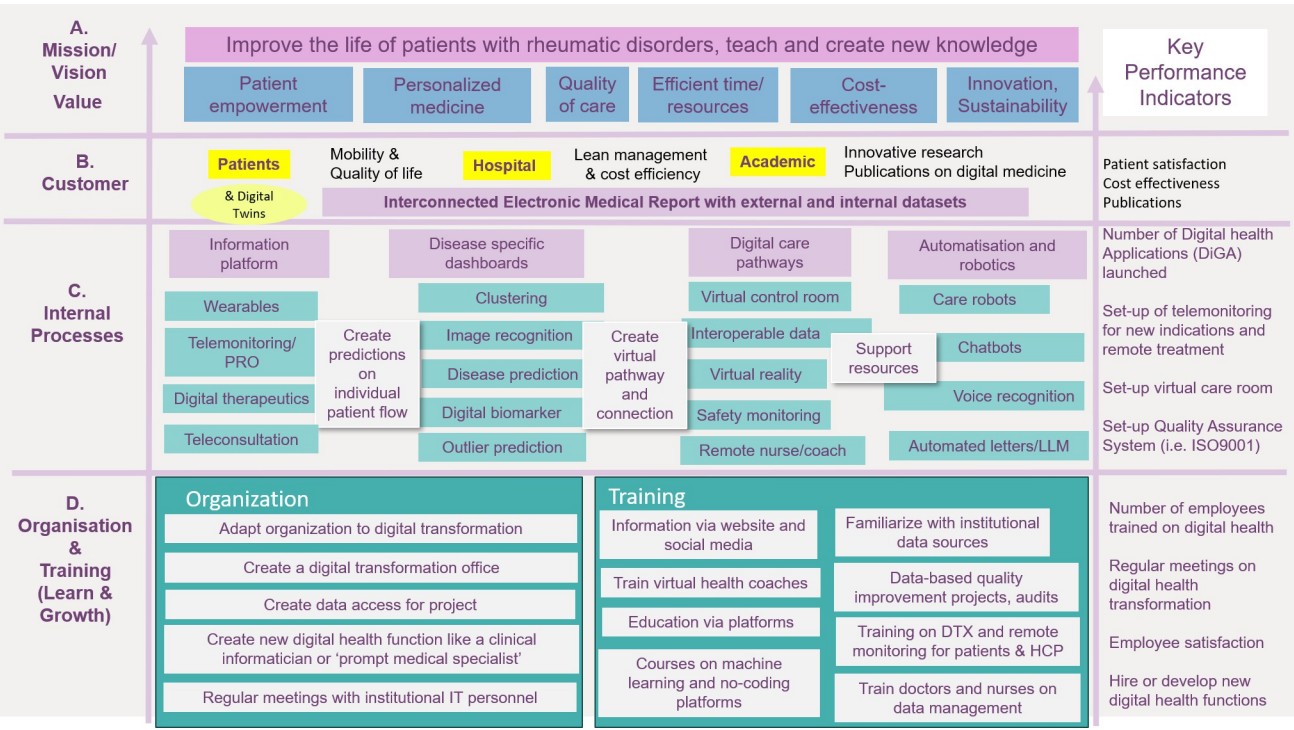

**Fig 2. A strategic map based on the Balanced Scorecard for the implementation of digital health in a rheumatology department in an academic center.** From bottom-up the figure shows people & organisation, internal processes, customers, values and missions. Values have been adapted to current healthcare trends. Exemplary indicators are listed on the right. Artificial intelligence based models are indicated in orange. DTX: Digital therapeutics. PRO: Patient reported outcomes. EMR: Electronic medical record.

## Values

Apart from a financial aspect, we defined four health care developments as the key values in our department:

*1.* Patient empowerment and convenience

Healthcare is developing towards a consumer-driven service. Patients want to actively chose their site of patient care and doctors. Less wating times, accessiblity via email ect. appointments online ect. This also applies at least for a portion of the patients that prefer a more active role in managing their chronic disease by monitoring their symptoms, providing patient reported outcomes (PROs) or even taking over clinical decisions after appropriate training [8]. Patient convenience from the medical and digital side (not in terms of hospitality) has so far remained largely unexamined in hospitals, although it has a significant impact on the quality of life as a measurable indicator. Of note, convencience can be more important to patients than quality of care [9].

*2. Personalized medicine*

Personalized medicine approaches aim to streamline individualized diagnostic and thera-peutic procedures for high efficacy and safety of care. This means a higher diagnostic effort (e.g. by -omics) and the integration of digital biomarkers and real world data in our workflow [10]. The exploitation of biomedical and clinical data by machine learning algorithms allows the prediction of individual disease courses, treatment response or phenotyping (clustering) as clinical decision support systems [11]. Thus, higher costs in diagnostics and data analysis might be compensated by better efficiency, lower complications and secondary healthcare costs.

*3. Interdisciplinary care*

In many chronic diseases, notably immune-mediated diseases or cancer, symptoms of dif-ferent body systems often are compulsorily connected e.g. skin and joint inflammation in pso-riasis arthritis, immune dysregulation due to immunotherapy in cancer or chronic pain and depression [12]. Quality of care depends on an optimal exchange of data and clinical decision between medical and paramedical information which bears a substantial number of barriers in terms of data interoperability. The individual 'point of care' which is not always clear to the patient (private practice, hospital, pharmacy, online), needs to be redefined. Digital platforms or EMR with interdisciplinary dashboards may support an efficient and time saving exchange [13].

*4. Sustainability and automatisation*

The shortage of healthcare professionals, notably nurses, doctors or administrative staff is an emerging problem which leads to employee dissatisfaction and sick leave. Automatisation of simple procedures such as voice, image or text recognition has largely been implemented e.g. for medical documentation. There are several FDA-approved algorithms for automated evaluation of radiographs e.g. for fracture detection on the market. So far, those solutions are not an active clinical decision support but they likely increase quality as 'double check' or if no radiologist is available. Finally, advanced technology in chatbots such as ChatGPT from OpenAI could potentially support administrative tasks such as writing and correcting medical reports. Care robots are slowly touching ground in hospitals and care homes in order for trans-port but also for vital signs and PRO assessment [14].

## Customers

Customers of our structure are patients (whether face-to-face or remote), clinicians, healthcare professionals, administrative staff and scientists. Accordingly, the needs of those internal and external customers on the BSC are anchored as clinical decision support, patient support, administrative support and research infrastructure. New types of customers are entering healthcare systems such as remote healthcare professionals (e.g. telenurse or online coaches), DIGA (Digital health application) providers ect. [15].

## Internal processes (Digital Care Pathways)

Internal processes in hospitals are responsible to maintain and improve quality, efficiency and safety. Fig 2 (left) shows various forms of digital data collection (telemonitoring, digital bio-markers, biosensors ect.) as the basis for a personalized medicine. AI-models likely integrated in the EMR permit to define phenotypes, disease predictions and to perform transfer learning from concomitant (autoimmune) diseases. This supports the creation of digital care pathways that organise individual patient monitoring and treatment and orchestrate ressouces optimally. Data accessibility and connectibility is a main process for horizontal digitalization, both in terms of providing user interfaces, data privacy and legal certainty. Usability and intepretability of data is considered as a key process to avoid data and knowledge silos. This concerns structured data at the moment, but can be enlarged unstructured data e.g. by large language models. Clearly, external registries or other data sources should be interoperable with hospital systems, especially the EMR.

Generally, the EMR can be considered as the main tool to integrate new digital solutions horizontally and vertically (Fig 3). For example. EMR can be extended vertically by integrating PROs, Patient reported experiences (PREs), digital biomarker or apps [16]. According to the 'Internet of Things' (IoT), the concept 'Internet of Medical Devices' (IoMD) should be promoted. Horizontally, dashboards increase usability of EMR between specialities and AI-generated predictions (including transfer learning algorithms) can be included in those interdisciplinary dashboards. New generation clinical dashboards also should show a holistic patient journey including predictions and outlier analysis etc. In the sense of horizontal integration along the value chain, the prediction should be connected to data on the availability of follow-up care facilities. The first step of AI-guided clinical pathway / decision support will likely be auto-care loops in stable disease courses. A maximal amount of collected data will be analyzed by clustering and prediction models, a digital treatment plan will be established and

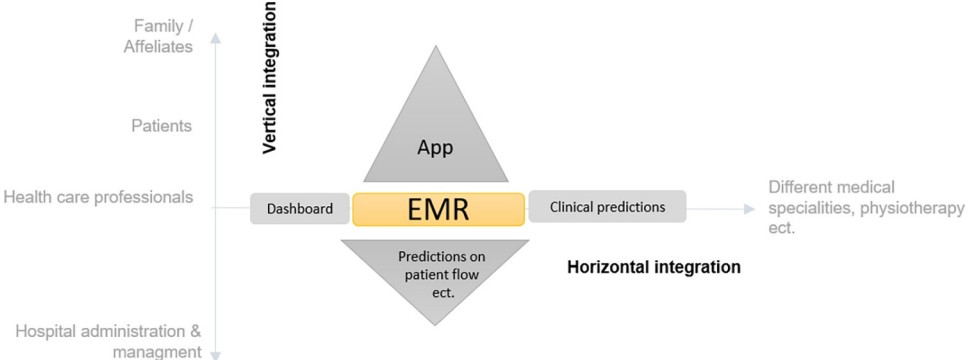

**Fig 3. Horizontal and vertical extension of Electronic Medical Records (EMR).** Elements around the EMR have been taken from the Balanced Scorecard (internal procedure perspective) and regrouped for a better demonstration.

automatically monitored and reported to health care professionals including remote nurses. Smart agenda planning allows to identify and planify patients with a high flare risk for a conventional visit.

Finally, automatisation is a sine qua non to encounter the shortage of rheumatologists and health care professionals. Large language models (LLM) such as GPT4 can be used to screen EMR and hospital databases for data or for the generation of discharge letters [17]. Voice recognition systems have widely been introduced in the clinic. New generation chatbots are able to extract the necessary information to create discharge reports and thus save time for the clinician to spend more time with patients or in interdisciplinary discussion. It seems primordial that all staff are equipped with smartphones and apps for those dashboards. Communication between colleagues in- and outside the hospital is via Email, which seems not adequate for several reasons (high number of unreplied messages, spam, security ect). It is obvious to connect case-based communication to dashboards in order to save time and to get relevant Information in a few clicks. All those digital processes together with their KPIs illustrate digital care pathways.

## People and organisation (learning organisation)

In order to implement (digital or other) innovations, the human factor remains indispensable. This is especially true in medicine, where human interaction and ethical concerns are or paramount importance. Therefore, change management is the basis for creating readiness and acceptance for new technologies. In chronic diseases, a holistic view on patient journeys (or digital clinical pathways) with patient-centered endpoints and indicators are of increasing importance. Both patients and healthcare professionals need to understand how to access and inteprete this data and how to guarantee data privacy. To this end, knowledge about the inhouse databases (data warehouse etc.) and external data sources (registries or apps) as well as basic knowledge on data science is required and should be educated as a first step. The national or federal strategy for education of digital health is currently being actively discussed, but is only interoperable to a limited extent due to the different health systems [18]. Thus, a regular exchange between institutional, pan-departmental data scientists and collaborators within clinical departments 'closer to clinical action', shoud be established. Clinical-informaticians within medical departments are key elements for horizontal digitalization. For exemple they can educate clinicians use EMR more efficiently, including processes such as billing or implementing new processes such as prescribing DIGAs (Digitale Gesundheitsanwendung, digital health application) [15].

Human clinical know-how is also required to build and to control machine learning algorithms. As for input variables for algorithms (e.g. for clinical predictions), it is obviously not just about the availability but also the quality of the clinical data (e.g. what, when and how was measured). Clinicians need to specify the relevance of the data. As an exemple in image recognition, clinically defined pre-processing increases the prediction performance. As an example focusing on the hip shape to predict radiographic progression of hip ostearthritis [19].

No-coding platforms for image recognition or natural language processing can support clinicians in creating on predictions based on the insitutional data. For natural language processing, chatbots such as the openAI chatbot can leverage the work of doctors and scientists by support of writing articles or create automated reports and save precious time. Those tools are typically explored in defined research projects or audits before integrating them in the clinical workflow. Clearly, data safety issues have to be addressed rigorously and with new technological tools, its strengths and weaknesses are yet to be fully understood–and there appears early indication that such chatbots can be prone to confabulating information. An example of a medical report generated by the chatGPT3 chatbot can be found in S1 Table.

Patient education in the use of wearables, apps, digital therapeutics or biomarkers is pivotal. However low adherence to digital tools such as DIGAs is a widely unadressed problem [20]. Beside gamification, avatars or better human companion (e.g. health coaches) are necesary to assure adherence and quality of care. For mental health care, which also affects a substantial part of patients with immune-mediated diseases in form of secondary fibromyalgia, the human companionship seems even more important. Therefore, the need for trained remote health care professionals, e.g. as health coaches, will increase strongly in the future. A new profession of online nurses is also emerging among nurses, who work for hospitals or insurance companies and are connected to patients via a wide range of digital tools [21].

On a national and internationa level, both societies such as EULAR and patient organisations foster education and exchange of digital solutions [22]. To support this endavour, we created the digital rheumatology network (www.digitalrheumatology.org) organising yearly conferences (Digital Rheumatology Days) and regular pod-and webcasts.

## Indicators

The BSC has been developed around key indicators so they can and should be obtained at all levels. In other words, what is measurable within a digital care pathway should be measured. In our opinion, extra hours, sick leave and employer satisfaction are among the most urgent KPI, given the shortness of health care professionals and the wish for part-time positions. Patient empowerment and convenience measured by patient satifaction is also notably important to create and maintain trust as the basis of successful care.

For skills, education levels both of patients and clinicians can be measured by the number of attended courses, podcasts ect. Prescribed DIGAs and telemonitoring reports and patient adherence to those can be measured and discussed during face-to-face consultatons. PREs can be used to reflect the mix of human and digital therapeutic services offered by the department. Internal processes, including automatisation procedures, can be monitored by reduced extra-hours. Measuring 'connected care' is more subtle but could be measurable by reduced length of hospital stay or readmission rates and patient satisfaction. Of note, many aspects of patient care such as empathy, time for listening can not be measured.

## Implementation in our department and first results

In our and many other hospitals, the key aspects of digital transformation are initiated top-down from an institutional level. For example, the online assignment of patients for consultations or hospitalisation or telemonitoring via an app for post-operative patients that is supervised by a central nurse team. The access to the data warehouse for scientific purposes has been facilitated and a machine learning team for clinical predictions or biomedical (big) data analysis collaborates closely with medical departments on a project level. To leverage these and other opportunities, we initiated a bottom-up strategy according to the above mentioned factors. On a departimental level, we have introduced regularly scheduled meetings between clinicians and institutional IT specialists. A clinician-informatician consultant provides regular training in basic machine learning coding and institutional data access to our clinical and research team. Clinical data science training using data from registries and our datawarehouse has been added to the curriculum for new assistant doctors. A specialized nurse consultation for rheumatic patients has been implemented to instruct patients in the use of apps with PROs, here within our national registry SCQM (Swiss Cohort for Quality Management), wearable data and the use of digital therapeutics. As our current EMR is lacking clinical scores and indicators such as the DAS28 score, we included the mannequin for tender and swollen joints and were thus able to synchonize EMR with the registry. Patients can send pictures or videos of

their hands into our system via apps. We have developed an algorithms that automatically measure the finger folds, joint diameter etc. on hand images to monitor joint swelling [23]. The range of motion can be estimated on self-recorded videos of patients by automatically measuring the angles of the different joints and comparing them with the previous values. In the future, hopefully such disease-specific biomarker will earlier detect of arthritis flares.

A telemonitoring & communication office is currently being set up where medical medical assistants are trained to become 'clinical workflow & communication manager' with access to PROs, wearables, photos, videos etc. on specific dashboards and obviously the agenda of our consultation. In parallel to telephone calls and emails from patients they forward information (or not) to specialized nurses or our rheumatologist and to refer patients to the consultation.

Five years ago we started an international conference series called the 'Digital Rheumatology Days' (www.digitalrheumatology.org) for healthcare professionals to educate them on telemonitoring, digital therapeutics/DIGAs, digital biomarkers, social media usage, machine learning ect. The annual conference was complemented by the Digital Rheumatology Network as an educational platform where information, podcasts, webcasts, etc. are permanently published and distributed via social media. To foster interdisciplinarity as part of the connected care model, the Common Ground Meeting for Immune-mediated Diseases has been created (common-ground-meeting.org), assembling different specialties for disease updates, the use of data (e.g. presentation of digital biomarkers) and case discussions by rheumatologists, dermatologists, gastro-enterologists, immunologists, nephrologists and pneumologists.

So far, the internal processes mentioned above have been partially implemented. Digital tools and data science have mainly been introduced on a project basis, with ongoing studies (S1 Table).

## Discussion

We present a comprehensive mapping of digital transformation in a rheumatology hospital department, focusing on current digital developments and the evolving values in healthcare. By utilizing the BSC framework, we effectively illustrate how the implementation of digital solutions, both existing and upcoming, can enhance our performance on different levels towards achieving a 'connected care model' that aligns with the institution's mission and vision.

First of all, this strategic mapping emphasizes the significance of bottom-up education for all stakeholders, including patients, to promote the adoption of new digital tools. We propose that knowledge on data science, app technology, cloud computing, and the fundamentals of AI, such as large language models, is a vital process in transforming healthcare organizations into learning environments that better support patients, administration, and healthcare professionals. In this context, the BSC may improve organizational performance by facilitating double-loop learning and disseminating the hospital's vision as a learning organization [24].

During our research, we identified a knowledge gap concerning institutional digital solutions or collaborative projects developed within the hospital. For instance, a platform designed for accessing clinical data for research purposes faced low adoption rates primarily due to a lack of awareness among clinical departments. Regular meetings between the institution and clinicians or the appointment of a clinician-informatician emerged as crucial strategies to leverage digitalization and foster adoption.

Furthermore, we demonstrate that the electronic EMR plays a pivotal role in the digital transformation of healthcare. As illustrated in Fig 3, The EMR can be expanded both horizontally and vertically e.g. through the integration of AI algorithms, apps, interdisciplinary dashboards, wearable data, patient-reported outcomes (PROs), ultimately serving as the central point of care within a hospital.

Initially, the BSC was developed to establish relationships between actions and process performance in business institutions at levels beyond financial values. The hospital as 'institution' has evolving values and a very short half-life of knowledge and technology.

In a first step, we established a BSC framework that corresponds to the current state of knowledge and digitalization. Over time, key success factors in hospitals have undergone changes, particularly in terms of employee satisfaction and patient empowerment. Interdependencies with other sub-units (= interdisciplinarity) have also become increasingly important and should be taken into account when selecting KPI.

It is important to note that this strategic mapping does not serve as a manual for digitalization but rather aims to inspire others to conduct similar exercises in order to create a 'compass' for responsible healthcare professionals.

We must acknowledge several limitations of this work. KPI implementation was only partial, and processes at different levels have not yet been fully connected or supported by empirical data. Thus, we cannot demonstrate causal relationships among key indicators within or across the four perspectives, as previously shown in hospital settings [25]. Therefore, the full potential of the BSC remains untapped. Adoption of the BSC healthcare professionals may prove challenging [5]. Another limitation lies in the fact that the BSC has shown limitations in measuring human relations, which are crucial components of learning organizations in the healthcare sector [26].

In conclusion, digital transformation is a complex and ongoing process that requires meticulous planning on various levels. The BSC proves to be a suitable tool for both planning and monitoring this significant endeavor. Digital tools should enable better care and save time, allowing healthcare professionals to focus on valuable human interaction that algorithms can never replace.

## Supporting information

**S1 Table. Implemented digital aspects in our rheumatology department.**
(DOCX)

## Acknowledgments

We thank Dr. Chris Lovejoy for his valuable review of the manuscript.

## Author Contributions

**Conceptualization:** Thomas Hügle, Vincent Grek.

**Data curation:** Thomas Hügle, Vincent Grek.

**Formal analysis:** Thomas Hügle, Vincent Grek.

**Methodology:** Thomas Hügle, Vincent Grek.

**Software:** Thomas Hügle, Vincent Grek.

**Visualization:** Thomas Hügle.

**Writing – original draft:** Thomas Hügle.

**Writing – review & editing:** Thomas Hügle, Vincent Grek.

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
