## [Decision Letter · Decision Letter 0]

4 Apr 2023

PDIG-D-23-00034

Horizontal Digitalization in Hospitals: a Roadmap towards Connected Care

PLOS Digital Health

Dear Dr. Hügle,

Thank you for submitting your manuscript to PLOS Digital Health. After careful consideration, we feel that it has merit but does not fully meet PLOS Digital Health's publication criteria as it currently stands. Therefore, we invite you to submit a revised version of the manuscript that addresses the points raised during the review process.

Please submit your revised manuscript within 60 days Jun 03 2023 11:59PM. If you will need more time than this to complete your revisions, please reply to this message or contact the journal office at digitalhealth@plos.org. Please include the following items when submitting your revised manuscript:

We look forward to receiving your revised manuscript.

Kind regards,

Benjamin P. Geisler, M.D., M.P.H., F.A.C.P., M.R.C.P. (London), F.H.M.

Academic Editor

PLOS Digital Health

Journal Requirements:

1. Please provide separate figure files in .tif or .eps format only and remove any figures embedded in your manuscript file. Please also ensure that all files are under our size limit of 10MB.

2. We noticed that you used "unpublished" in the manuscript. We do not allow these references, as the PLOS data access policy requires that all data be either published with the manuscript or made available in a publicly accessible database. Please amend the supplementary material to include the referenced data or remove the references.

Additional Editor Comments (if provided):

Although the reviewers found the manuscript promising, it is not quite as organized as we need it to to both be easily digestible for the reader and to conform with PLOS requirements. Specifically, I would discourage the use of the numbered list in the abstract. Also, per PLOS authors' guidelines, "While the Abstract is conceptually divided into three sections (Background, Methodology/Principal Findings, and Conclusions/Significance), do not apply these distinct headings to the Abstract within the article file." Similarly, it might not be wise to make the discussion long and discuss the four issues that have separate headlines (some of this might be more for the discussion). The end of the introduction lacks a clear goal/objective. For the methods & materials, consider using more references if you don't want to spell out your methods (e.g., software used, prior categorization etc) too much. A part of the results likewise contains elements that normally appear in the discussion, e.g., that you plan to use ChatGPT next. The entire results section might benefit from a reorganization, so that the reader can follow more easily. Results don't usually contain bullet lists; consider a table for it. The discussion should start with a high-level summary of your results. Remember that parts of the introduction and results should be moved to the discussion, as appropriate. My main criticism is that at the end of the discussion, what appears to be the conclusion is far too general; try to put in the implications here - should other implement BSC approaches, or is it too early (or would you recommend against it)? What should be kept in mind, and what needs to be researched further? We look forward to your revision.

Reviewers' comments:

Reviewer's Responses to Questions

**Comments to the Author**

1. Does this manuscript meet PLOS Digital Health’s publication criteria? Is the manuscript technically sound, and do the data support the conclusions? The manuscript must describe methodologically and ethically rigorous research with conclusions that are appropriately drawn based on the data presented.

Reviewer #1: No

Reviewer #2: No

Reviewer #3: Yes

Reviewer #4: Yes

Reviewer #5: Partly

2. Has the statistical analysis been performed appropriately and rigorously?

Reviewer #1: N/A

Reviewer #2: N/A

Reviewer #3: N/A

Reviewer #4: No

Reviewer #5: N/A

3. Have the authors made all data underlying the findings in their manuscript fully available (please refer to the Data Availability Statement at the start of the manuscript PDF file)?

Reviewer #1: Yes

Reviewer #2: Yes

Reviewer #3: Yes

Reviewer #4: No

Reviewer #5: Yes

4. Is the manuscript presented in an intelligible fashion and written in standard English?

Reviewer #1: No

Reviewer #2: Yes

Reviewer #3: Yes

Reviewer #4: Yes

Reviewer #5: Yes

5. Review Comments to the Author

Reviewer #1: Thank you for this opportunity to review this manuscript.

This manuscript is an opinion. It aims to present a scalable and understandable matrix to instruct and foster

horizontal digital transformation in hospitals from a medical side.

However, I think that major revisions have to be made to this manuscript before publishing it in PLOS Digital Health. First, the introduction has to be re-written with a clear definition of the horizontal digital transformation. The Balanced scorecard (BSC) has to be defined more clearly in the introduction. The aim does not mention the utilization of BSC. It is not clear how the new opportunities were connected with BSC or the methodology. The importance and knowledge gap of this manuscript are not presented. Why did the authors wrote this manuscript? what are the practical implications? 

Regarding the methodology, it lacks a description of the time frame of implementation. Also it is not clear how BSC is connected to it. Usually, the KPIs are designed by a panel or committee. The face and content validity, or the measurability and feasibility are assessed before implementation. Also, the figure used is not clear. the source of each KPI is not clear. The internal process perspective and knowledge and information perspective are mixed together.

Other minor comments: the language have to be revised. Examples: 1- suported instead of supported in the abstract, 2- ect instead of etc 3- undadressed problem instead of unaddressed. 

Also the authors referred to the four perspectives in the methodology as four key performance indicators. This has to be corrected. To understand the design of BSC further we recommend to have a look on the BSC perspectives, dimensions, and KPIs (https://doi.org/10.1186/s12913-022-07863-0).

Reviewer #2: How managing digital transformation implementation in healthcare is a timely challenge. Moreover, I found a promising case study described in this manuscript. However, I was overwhelmed by the rich flow of information and the manuscript in this release appears out-of-focus. I set below a list of recommendations that would assist the authors while revising their study.

1. What is the real aim of the study? The abstract does not mention it. In the introduction section you mention two aims, firstly “In this article, we aim to show how digital solutions respond to those clinical and administrative needs and what is needed to implement them in clinical practice”; secondly “This article aims to present a scalable and understandable matrix to instruct and foster horizontal digital transformation in hospitals from a medical side”. And the following sections describe much more than these two aims. Probably, your case study could provide data for more than one article, but each article must have a proper research question and the empirics will provide an answer to it. For example, in this article, you describe 1. causes of horizontal integration of digital solutions, 2. Solutions that would implement, 3. Solutions implemented, 4. Results of the implementation. In my opinion, you have to choose your focus and describe properly one of these points. The others could be at most briefly mentioned. You could revise entirely the manuscript according to the novel angle you choose to describe.

2. Why do you consider the balanced scorecard a proper framework to refer to your story? The Balanced Scorecard is a performance management system that aims to help the execution of strategy at the organizational level. Moreover, this system presents four perspectives and the key performance indicators identified in each of these perspectives present a cause-effect link. Thus, your use at the project level, with other perspectives and without a clear identification of leading and lagging indicators should be explained. The balanced scorecard is a powerful management system, but it is not a roadmap. It requires a careful design and implementation process and the creation of strategy maps. I suggest reading papers about balanced scorecard implementation and its use in organizational contexts. Hope this published paper could be of any help.

Tawse A., Tabesh P. (2023). Thirty years with the balanced scorecard: What we have learned. Business Horizons, 66(1), 123-132.

Ming-Chin Y., Yu-Chi T. (2006), Using Path Analysis to Examine Causal Relationships Among Balanced Scorecard Performance Indicators for General Hospitals. The Case of a Public Hospital System in Taiwan. Health Care Management Review, 31(4), 280-288.

Epstein M., Manzoni J-F (1997), The balanced scorecard and tableau de bord: Translating strategy into action. Management Accounting, 79(2), 28-36.

3. The methodological section is completely skipped. The methodological rigour in collecting and analysing data is very important. You have a good entrance on the case study and maybe you managed the entire transformation process. Hence, you could benefit from a lot of details: documents, notes of informal meetings, and interviews. If you actively participate in the implementation process, you could approach the action/interventionist research method. It requires adequate theoretical preparation to be distinguished from consultancy, but it could give you a lot of satisfaction. You could analyse these studies.

Baard, V. (2010), "A critical review of interventionist research", Qualitative Research in Accounting & Management, Vol. 7 No. 1, pp. 13-45. https://doi.org/10.1108/11766091011034262

Lukka K., Wouters M. (2022), Towards interventionist research with theoretical ambition, Management Accounting Research, 55, https://doi.org/10.1016/j.mar.2022.100783

Reviewer #3: This paper instructs and foster horizontal digital transformation in a Swiss academic rheumatology department, using a Balanced Scorecard approach.

The subject is very interesting and relevant.

The paper is well-written and structured. 

The approach of the balanced scorecard was adequate. The perspectives were well-adjusted to the case and aligned with the mission, vision, and values. 

I recommend removing abbreviations from the abstract (eg ERM).

Despite liking the title, I would consider replacing it and including the “balanced scorecard” in it (for example Horizontal Digitalization in Hospitals using Balanced Scorecard: a Roadmap towards Connected Care)

Reviewer #4: your research objective is to present the Case of your institution's works in digital transformation. Why should use the Balanced Scorecard to do this research? what is the relationship between the method results and the routine?where is the Data analysisi from BS method?

the paper should be based on the medical demand to put forward the routine and describe the details of that.

another the paper's format should be improved.

Reviewer #5: Dear Author, 

Thank you for your contribution. 

Although the paper focuses on a hot topic, there are several issues to improve.

The main limitation of the paper is the methodology, how did you collect data? how did you build the balanced scorecard? how did you identify the indicators?

Moreover there are other few comments:

- the abstract is quite longer than the limit fixed by the journal and the aim is not clear;

- the first part of the findings (until the implementation section) is not clear because of the methodology, it is unclear how you built the results by using the BSC.

Finally, please, check the english language.

6. PLOS authors have the option to publish the peer review history of their article (what does this mean?). If published, this will include your full peer review and any attached files.

**Do you want your identity to be public for this peer review?** For information about this choice, including consent withdrawal, please see our Privacy Policy.

Reviewer #1: Yes: Faten Amer

Reviewer #2: No

Reviewer #3: No

Reviewer #4: Yes: Xinli Zhang

Reviewer #5: No

---

## [Decision Letter · Decision Letter 1]

15 Aug 2023

PDIG-D-23-00034R1

Digital Transformation of an Academic Hospital Department: a Case Study on Strategic Planning Using the Balanced Scorecard

PLOS Digital Health

Dear Dr. Hügle,

Thank you for submitting your manuscript to PLOS Digital Health. After careful review, we recommend that you submit a revised version of the manuscript that addresses the points raised during the review process.

Please submit your revised manuscript within 30 days (on or before September 15, 2023). If you will need more time than this to complete your revisions, please reply to this message or contact the journal office at digitalhealth@plos.org. Please include the following items when submitting your revised manuscript:

We look forward to receiving your revised manuscript.

Kind regards,

Benjamin P. Geisler, M.D., M.P.H., F.A.C.P., M.R.C.P. (London), F.H.M.

Academic Editor

PLOS Digital Health

Journal Requirements:

Additional Editor Comments (if provided):

Reviewers' comments:

Reviewer's Responses to Questions

**Comments to the Author**

1. If the authors have adequately addressed your comments raised in a previous round of review and you feel that this manuscript is now acceptable for publication, you may indicate that here to bypass the “Comments to the Author” section, enter your conflict of interest statement in the “Confidential to Editor” section, and submit your "Accept" recommendation.

Reviewer #1: All comments have been addressed

Reviewer #2: All comments have been addressed

Reviewer #3: All comments have been addressed

2. Does this manuscript meet PLOS Digital Health’s publication criteria? Is the manuscript technically sound, and do the data support the conclusions? The manuscript must describe methodologically and ethically rigorous research with conclusions that are appropriately drawn based on the data presented.

Reviewer #1: Yes

Reviewer #2: Yes

Reviewer #3: Yes

3. Has the statistical analysis been performed appropriately and rigorously?

Reviewer #1: Yes

Reviewer #2: N/A

Reviewer #3: N/A

4. Have the authors made all data underlying the findings in their manuscript fully available (please refer to the Data Availability Statement at the start of the manuscript PDF file)?

Reviewer #1: Yes

Reviewer #2: Yes

Reviewer #3: Yes

5. Is the manuscript presented in an intelligible fashion and written in standard English?

Reviewer #1: Yes

Reviewer #2: Yes

Reviewer #3: Yes

6. Review Comments to the Author

Reviewer #1: I would like to thank the authors who took all the editor and reviewers' comments into consideration. 

I believe that the quality of the manuscript had improved a lot.

Just a minor comment regarding the revised manuscript: Please notice that some words are defined more than once such as Key Performance Indicators. Please make sure to define only one time at the first mention.

I recommend the last version for publication

Reviewer #2: The auditors did a great job reviewing the article. In my view, it could be accepted in its present form.

Reviewer #3: Thank you for addressing my comments.

7. PLOS authors have the option to publish the peer review history of their article (what does this mean?). If published, this will include your full peer review and any attached files.

**Do you want your identity to be public for this peer review?** For information about this choice, including consent withdrawal, please see our Privacy Policy. 

Reviewer #1: Yes: Faten Amer

Reviewer #2: No

Reviewer #3: No

---

## [Editor Report · Decision Letter 2]

10 Oct 2023

Digital Transformation of an Academic Hospital Department: a Case Study on Strategic Planning Using the Balanced Scorecard

PDIG-D-23-00034R2

Dear Prof. Hügle,

We are pleased to inform you that your manuscript 'Digital Transformation of an Academic Hospital Department: a Case Study on Strategic Planning Using the Balanced Scorecard' has been provisionally accepted for publication in PLOS Digital Health.

Best regards,

Benjamin P. Geisler, M.D., M.P.H., F.A.C.P., M.R.C.P. (London), F.H.M.

Academic Editor

PLOS Digital Health